# Unveiling Moroccan Nature’s Arsenal: A Computational Molecular Docking, Density Functional Theory, and Molecular Dynamics Study of Natural Compounds against Drug-Resistant Fungal Infections

**DOI:** 10.3390/ph17070886

**Published:** 2024-07-04

**Authors:** Imane Yamari, Oussama Abchir, Hassan Nour, Meriem Khedraoui, Bouchra Rossafi, Abdelkbir Errougui, Mohammed Talbi, Abdelouahid Samadi, MHammed El Kouali, Samir Chtita

**Affiliations:** 1Faculty of Sciences Ben M’Sik, Hassan II University of Casablanca, Sidi Othman, Casablanca P.O. Box 7955, Morocco; yamariimane86@gmail.com (I.Y.); oussamaabchir12@gmail.com (O.A.); meriemkhedraoui5@gmail.com (M.K.); bouchrarossafi7@gmail.com (B.R.); a_errougui@yahoo.fr (A.E.); talbi.uh2c@gmail.com (M.T.); m.elkouali@gmail.com (M.E.K.); 2Department of Chemistry, College of Science, United Arab Emirates University, Al Ain P.O. Box 15551, United Arab Emirates

**Keywords:** antifungal activity, molecular docking, dynamics simulation, Density Functional Theory, ADME-Toxicity

## Abstract

*Candida albicans* and *Aspergillus fumigatus* are recognized as significant fungal pathogens, responsible for various human infections. The rapid emergence of drug-resistant strains among these fungi requires the identification and development of innovative antifungal therapies. We undertook a comprehensive screening of 297 naturally occurring compounds to address this challenge. Using computational docking techniques, we systematically analyzed the binding affinity of each compound to key proteins from *Candida albicans* (PDB ID: 1EAG) and *Aspergillus fumigatus* (PDB ID: 3DJE). This rigorous in silico examination aimed to unveil compounds that could potentially inhibit the activity of these fungal infections. This was followed by an ADMET analysis of the top-ranked compound, providing valuable insights into the pharmacokinetic properties and potential toxicological profiles. To further validate our findings, the molecular reactivity and stability were computed using the DFT calculation and molecular dynamics simulation, providing a deeper understanding of the stability and behavior of the top-ranking compounds in a biological environment. The outcomes of our study identified a subset of natural compounds that, based on our analysis, demonstrate notable potential as antifungal candidates. With further experimental validation, these compounds could pave the way for new therapeutic strategies against drug-resistant fungal pathogens.

## 1. Introduction

Fungal infections, while historically underestimated in the broader spectrum of infectious diseases, have emerged as significant health threats, particularly in immunocompromised populations [1]. As eukaryotic microorganisms, fungi display complex cellular structures and biological processes contributing to host survival and pathogenicity. Unlike bacteria, fungal cells have rigid cell walls composed of chitin and β-glucans, which not only provide structural integrity but also play roles in immune evasion [2]. The pathogenic potential of fungi is attributed to a suite of virulent factors. These include the ability to switch between yeast and hyphal forms, known as dimorphism, which is particularly prevalent in *Candida* species and aids in tissue invasion [3,4]. Additionally, some fungi produce mycotoxin-potent compounds that can impair host immune function, damage tissues, or even lead to carcinogenic effects in long-term exposures [5]. Biofilm formation is another crucial virulence mechanism. When fungal cells aggregate and form biofilms on surfaces, they exhibit increased resistance to antifungal agents and contribute to persistent infections [6,7]. Drug resistance arises due to several factors. Genetic mutations can lead to changes in the target site of the drug, reducing its efficacy. Horizontal gene transfer allows bacteria to acquire resistance genes from other bacteria. Furthermore, the misuse and overuse of antibiotics in both human medicine and agriculture contribute significantly to the development and spread of resistant strains [8]. Among the myriad of fungal pathogens, *Candida albicans* and *Aspergillus fumigatus* have been particularly noteworthy, with both species presenting significant clinical challenges. *Candida albicans*, an essential and ubiquitous component of the human microbiota, can become an opportunistic pathogen causing infections ranging from superficial mucosal infections to life-threatening systematic candidiasis [9,10]. On the other hand, *Aspergillus fumigatus* is primarily known for causing invasive aspergillosis, especially in patients with compromised immune systems, such as those undergoing organ transplantation or chemotherapy [11]. The therapeutic landscape for fungal infections has been relatively stagnant over the past few decades, with only a handful of drug classes available. A major complicating factor in the treatment regimen is the emergence of drug-resistant strains [8]. Given this backdrop, there is an urgent need to explore alternative antifungal strategies. Natural compounds, with their vast structural diversity, have been a treasure trove in the quest for novel therapeutics [12,13,14,15]. Historically, several frontline antifungal agents, like griseofulvin and amphotericin B, are derivatives of naturally occurring compounds [16]. Recent investigations have shed light on the potential of several natural compounds demonstrating antifungal activity against drug-resistant strains; possible mechanisms include disrupting the fungal cell membrane, inhibiting essential enzymes involved in cell wall synthesis, and generating reactive oxygen species that induce cellular damage [17,18,19,20]. They have long been recognized for their immunomodulatory and antimicrobial properties. Compounds such as curcumin, resveratrol, and echinacea extract can modulate the immune response, enhancing the body’s ability to fight infections. Additionally, natural antimicrobials like tea tree oil, garlic, and honey exhibit broad-spectrum activity against various pathogens, including drug-resistant strains. Their mechanisms include disrupting microbial cell membranes, inhibiting biofilm formation, and enhancing host immune function [21,22]. Such findings underline the importance of systemically exploring nature’s repertoire to address the pressing challenge of fungal drug resistance. The current study is rooted in a critical juncture of two paradigms, the pressing need for novel antifungal agents and the revolutionary capabilities of computational methodologies. It presents a novel approach to identifying and optimizing antimicrobial agents using advanced computational techniques. The integration of natural and synthetic molecules in the design process represents a significant advancement in the field of drug discovery. With these advancements, we can now simulate, predict, and analyze molecular interactions at an incredible scale and speed [23,24,25]. Furthermore, by coupling the screenings with other computational tools, such as ADMET prediction and molecular dynamic simulations, we obtained a holistic understanding of the compound’s pharmacological profile, which not only includes its potential efficacy but also its pharmacokinetics, potential toxicities, and stability in biological systems. The findings highlight the potential of these ligands in developing new therapies against drug-resistant infections. In light of these capabilities, our study leverages the power of computational tools to delve deep into a collection of natural compounds, seeking those with promising antifungal activity against drug-resistant *Candida albicans* and *Aspergillus fumigatus*.

## 2. Results and Discussion

### 2.1. Molecular Docking

Before initiating the main docking experiments, a critical preliminary step involved conducting redocking simulations for the co-crystallized ligands: A70 in protein 1EAG and FAD in protein 3DJE. The outcomes demonstrated remarkable accuracy, with low RMSD values of 0.8 Å and 2 Å, respectively (Figure 1). These findings strongly affirm the reliability of the employed docking protocol in our study.

After that, we assessed the antifungal potential of 297 natural compounds against *Candida albicans* and *Aspergillus fumigatus*. The obtained binding energy scores shed light on the strength of interactions between the tested compounds and their respective fungal proteins. A comparative analysis with the reference antifungal drug, fluconazole, is presented in the table below. Negative binding energy scores indicate the stability of ligand–protein complexes, with lower scores denoting stronger interactions. Notably, our study focused on the top 40 compounds selected from the initial list of 297, all exhibiting a binding energy below −7 kcal/mol. These compounds emerge as promising antifungal agents, with a focus on multi-target activity against both *Candida albicans* and *Aspergillus fumigatus*, demonstrating a wide-spectrum effect on fungi. Among them, L1 (beta-glycyrrhetinic acid) from *Glycyrrhiza glabra* displayed a remarkable score of −9.6 kcal/mol, emphasizing its efficacy against *Candida albicans*. Additionally, L13 (liquiritin) from the same species, *Glycyrrhiza glabra*, achieved a remarkable score of −10.1 kcal/mol, demonstrating its effectiveness against *Aspergillus fumigatus*. The selection of compounds not only surpasses fluconazole but also highlights their diverse pharmacophores. These include hydroxyl groups for hydrogen bonding, aromatic rings for π-π stacking, nitrogen-containing heterocycles for binding specificity, and alkyl side chains for membrane interactions. These features contribute to their broad-spectrum antifungal activity, crucial for addressing diverse fungal pathogens. The data underline the significant value of these natural compounds, particularly L1 and L13, as candidates for further exploitation in the development of new, highly effective antifungal therapies.

The comprehensive results for ligands L1 to L40 and their respective scores for both fungal strains are detailed in the provided Table 1, and the 2D structures are included in the Appendix A.

Upon examining the interactions that resulted from the simulation, a comprehensive understanding of the binding modes within the protein’s active site was attained; the detailed analysis is depicted in Figure 2. For *Candida albicans*, the L1-1EAG complex revealed the establishment of two hydrogen bonds with crucial residues, including GLN54 (2.42 Å), ASP120 (2.04 Å), and a carbon–hydrogen bond with GLY220 (3.09 Å), along with a hydrophobic pi–sigma interaction with TYR84 (3.70 Å) that occurs between the carbon atom on the ligand L1 and the sigma electrons of the aromatic ring on TYR84. In contrast, the reference drug–1EAG complex exhibited a diverse interaction profile, involving additional hydrogen bonds, halogen bonds, and hydrophobic pi–sigma- and pi–pi-stacked interactions with specific residues GLY85 (2.99 Å), GLY220 (3.44 Å), ASP218 (3.65 Å), ASP86 (3.06 Å), THR221 (3.97 Å), TYR225 (5.10 Å), and TYR84 (4.51 Å).

For the *Aspergillus fumigatus* protein (Figure 3), the L13-3DJE complex exhibited substantial conventional hydrogen bonds with PHE367 (2.25 Å), formed between the hydrogen on PHE367 and the oxygen on the ligand (H-acceptor). Further, hydrophobic pi–sigma interaction was observed between the carbon atom of ALA47 (3.74 Å) and the sigma electrons of the aromatic ring (pi-orbitals). Pi–alkyl bonds were formed between the ligand’s aromatic system and the alkyl side chain of residues ALA362 (4.71 Å), CYS337 (4.58 Å), and LYS53 (4.59 Å). Conversely, the reference drug–3DJE complex displayed a diverse interaction repertoire, including conventional hydrogen bonds with ARG112 (2.54 Å), LYS368 (2.49 Å), GLU280 (3.79 Å), GLY364 (3.34 Å), ARG343 (3.85 Å), and hydrophobic pi–sigma- and pi–pi-stacked interactions with TRP236 (3.70 Å), ALA362 (4.78 Å), and others.

These findings underscore the specificity and versatility of ligand–protein interactions, contributing to our understanding of the molecular mechanism which provides a foundation for further investigations into potential therapeutic agents targeting these proteins. Molecular docking serves as a valuable tool, offering insights into how proteins interact with ligands at the molecular level. However, it is crucial to validate these computational predictions with experimental techniques such as Surface Plasmon Resonance (SPR).

### 2.2. Computational ADME-Tox and Drug-Likeness

In our investigation using the Swiss ADME web server, we conducted a thorough analysis of the bioavailability profiles of two compounds, L1 and L13, as shown in Table 2, aiming to conclude potential drug development.

Notably, L1, characterized by higher molecular weight and lipophilicity, exhibited a promising high gastrointestinal (GI) absorption potential, indicating its likelihood for effective absorption in the digestive tract. However, there is a violation of Lipinski’s rule of five, with a lipophilicity slightly greater than five, which is generally acceptable in many cases of drug development. On the other hand, L13, with a lower molecular weight and higher flexibility, presented a contrasting profile, predicting low GI absorption and offering potential advantages in terms of reduced molecular size and increased flexibility for enhanced tissue penetration. The higher polarity of L13, as indicated by a greater topological polar surface area, could facilitate interactions with biological targets. Both compounds displayed relatively high synthetic accessibility scores, indicating that the compounds are predicted to be accessible synthetically. The bioavailability scores provided a comparative measure, with L1 demonstrating a higher score than L13. These findings not only offer a comprehensive understanding of the molecular characteristics of L1 and L13 but also inform future decisions in the pursuit of effective drug candidates.

In Figure 4, our bioavailability radar provides a concise summary of the drug-likeness of the compounds L1 and L13. This radar plot incorporates six key physicochemical properties crucial for rapid drug-likeness assessment: lipophilicity, size, polarity, flexibility, and saturation [26]. The drug-like region is visually limited by a pink area on the radar plot. Notably, our compounds consistently reside entirely within this designated physicochemical range, underscoring their adherence to drug-like properties.

The pharmacokinetics and pharmacodynamics profiles of compounds L1 and L13 were assessed using the pkCSM web server (Table 3), which includes a set of essential ADME-Tox properties.

In terms of absorption, L1 demonstrated advantageous characteristics with higher water solubility, increased CaCO_2_ permeability, moderate skin permeability, and a notably higher human intestinal absorption percentage of 97.389%. Higher water solubility facilitates effective dissolution, while increased CaCO_2_ permeability suggests efficient absorption through the intestinal barrier. The substantial human intestinal absorption percentage further supports L1 potential as an orally administrated drug candidate. Additionally, L1 exhibited moderate skin permeability, a feature important for transdermal drug delivery. Furthermore, L1 suggested potential blood–brain penetration and central nervous system permeability, indicative of its potential efficacy in neurologically targeted applications. Distribution properties revealed that L1 exhibited a lower volume of distribution at a steady state, signifying a more controlled distribution within the body. For metabolism, L1 is metabolized by CYP3A4, a key liver enzyme involved in drug metabolism. Compound L13 exhibited moderate pharmacokinetic characteristics, including moderate water solubility, CaCO_2_ permeability, and human intestinal absorption. It displayed comparable skin permeability to L1, suggesting potential for transdermal delivery. The compound showed moderate distribution within the body, coupled with limited penetration through the blood–brain barrier and central nervous system, which can be advantageous, potentially minimizing off-target effects in neurological tissues. Metabolized by CYP3A4, L13 undergoes significant liver metabolism, a crucial consideration for drug–drug interactions.

The total clearance for both compounds, L1 and L13, provides insights into the rate at which these substances are eliminated from the body. For L1, the negative value indicates a moderately lower clearance rate. This may suggest a controlled elimination process, potentially leading to a more extended presence in the body, which can be advantageous for maintaining therapeutic concentration over time. Meanwhile, the L13 total clearance value was positive, indicating a relatively higher clearance rate, which suggests that the compound is eliminated from the body quickly. It may be desirable in certain cases to reduce the risk of prolonged drug exposure.

Notably, both L1 and L13 demonstrated negligible toxicity risks, as evidenced by the absence of AMES toxicity, hERG I inhibition, hepatoxicity, and skin sensitization. These comprehensive findings collectively suggest that L1 and L13, with their favorable pharmacokinetic attributes, are promising for further exploration as part of drug development efforts.

A more thorough evaluation through further testing is essential to elucidate the complete pharmacokinetic and pharmacodynamic profiles of compounds L1 and L13 and guarantee their safety and efficacy in potential therapeutic applications.

### 2.3. Molecular Quantum Analysis

The DFT calculations with the 6-31G (d,p) basis set have yielded important electronic and structural parameters for compounds L13 and L1 (Table 4).

Chemical parameters were computed using the following relationships: Ionization potential (I) was determined as −E_HOMO_, and electron affinity (A) as −E_LUMO_. Electronegativity (χ) was calculated as (I + A)/2, while chemical potential (μ) was derived as −(I + A)/2. Global hardness (η) was obtained as (I − A)/2, and global softness (σ) was defined as 1/η. The electrophilicity index (ω) was computed using the equation ω = μ^2^/(2η). The energy band gap (ΔE) between the HOMO and LUMO is a crucial indicator of the chemical reactivity and stability of compounds. Higher (ΔE) values, such as those observed for L13 (4.8885 eV) and L1 (4.8989 eV), suggest low chemical reactivity and high stability. These results indicate that both molecules are hard, exhibiting high kinetic stability and low chemical reactivity. This stability is supported by the negative values of the Pi chemical potential for both molecules (−3.7499 and −3.5528), indicating the spontaneity of the inclusion process.

The energy levels of HOMO and LUMO indicate that L13 has a greater ability to donate electrons (E_HOMO_ = −6.19413 eV) compared to L1 (−6.00229 eV). In comparison, L1 has a slightly lower energy for accepting electrons (E_LUMO_ = −1.10342 eV) compared to L13 (−1.30560 eV). Both compounds exhibit similar electrophilicity index (ω) values, representing a compound’s ability to acquire an electron charge, which is low for both molecules (2.876 for L13 and 2.577 for L1), suggesting they are less electrophilic. The electronegativity (χ) values for compounds L13 and L1 are 3.7498 and 3.5528, respectively, unveiling their affinities for attracting electrons. Both compounds exhibit closely aligned global hardness (η) and softness (σ) values, indicating a balanced resistance to changes in electron density. Notably, the dipole moment (μ) for L1 is substantial at 5.735663 Debye, signifying heightened charge separation, whereas L13 exhibits a lower dipole moment of 1.753767 Debye. Furthermore, the electronic energy values highlight the enhanced stability of L13 (−40,558.4) compared to L1 (−40,046.5).

#### 2.3.1. Frontier Molecular Orbitals (FMOs)

The analysis of boundary molecular orbitals, in particular the most occupied molecular orbital (HOMO) and the least occupied molecular orbital (LUMO), is a fundamental aspect of understanding the chemical reactivity and stability of compounds (Figure 5). In the context of L13 and L1 compounds, these orbitals play an essential role in understanding their electronic structures and potential interactions with other molecules. The HOMO is the highest-energy orbital containing electrons and primarily governs the compound ability to donate electrons during chemical reactions. The LUMO, on the other hand, is the lowest-energy orbital that contains no electrons; it indicates the compound ability to accept electrons. The energy difference between the most occupied molecular orbital (HOMO) and the least occupied molecular orbital (LUMO) is generally referred to as the energy band gap, denoted ΔE and calculated as ΔE = E_LUMO_ − E_HOMO_. This band gap is a crucial parameter in quantum mechanics and computational chemistry, as it indicates a molecule’s stability and reactivity.

For compound L13, the HOMO shows a primary location of the electron cloud, excluding the ring formed by atoms C20, C21, C22, C23, C24, and O25. Conversely, the electron cloud of the LUMO is distributed over half the molecule. This distribution suggests that the HOMO of L13 can donate an electron in the main region of the electron cloud, while the LUMO can gain an electron, indicating potential reactivity in half the molecule.

For compound L1, the HOMO is mainly located on the ring bearing the C=O group (between C14 and O26) and the directly bonded ring on its left side. The electron cloud of the LUMO is distributed over atoms O26, C13, and C14, in the region of the C21–C22 bond, and over the methyl group represented by atom C34. This distribution suggests that the HOMO of L1 can donate an electron in the region of the C=O ring and the adjacent ring, while the LUMO can gain an electron, particularly on atoms O26, C13, and C14, in the region of the C21–C22 bond, and on the methyl group represented by atom C34. These observations provide crucial insights into the potential reactivity of the L1 and L13 molecules.

#### 2.3.2. Molecular Electrostatic Potential (MEP)

The molecular electrostatic potential holds significant importance in predicting the chemical reactivity of compounds and understanding biological processes. It provides insights into the reactive sites of molecules, visually represented by the molecular electrostatic potential (MEP) surface through color coding. As illustrated in Figure 6, blue regions indicate positive potential values, signifying areas of low electron density and favoring nucleophilic attacks. Conversely, red regions represent negative potential values, suggesting sites prone to electrophilic attacks. Green regions indicate a neutral potential [27]. For molecule L13, the MEP analysis reveals a red-to-yellow region with negative potential concentrated around oxygen atoms, indicating these sites are favorable for electrophilic attacks. Most hydrogen atoms in the -OH group are susceptible to nucleophilic attacks due to the blue color around them, while green areas concentrate around carbon atoms, indicating a neutral potential. This description applies similarly to molecule L1.

### 2.4. Stability of Protein–Ligand Interactions MD Simulation Analysis

#### 2.4.1. Root-Mean-Square Deviation Analysis and the Root-Mean-Square Fluctuation

The root-mean-square deviation (RMSD), a crucial measure in molecular dynamics simulations, provides insight into the dynamic stability and structural changes of the complex during the 100 nanoseconds of simulation. As illustrated in Figure 7, the L13-3DJE complex revealed that the overall structural integrity of the protein backbone was well maintained throughout the simulation. The complete system exhibited a root-mean-square deviation of 4 Ǻ, indicating some flexibility or conformational changes in its position over time. Meanwhile, the protein backbone showed a modest root-mean-square deviation of 1.5 Ǻ, reflecting a relatively stable structure with slight deviations throughout the simulation. In the second L1-1EAG complex, the ligand exhibited a stable binding profile with a root-mean-square deviation (RMSD) of 1.8 Å, indicating a modest deviation from its initial conformation during simulation. At the same time, the protein showed a root-mean-square deviation of 1.2 Å, reflecting a similar stable structural profile. The close alignment of these RMSD values for the ligand and protein, both located in the same frame, underlines the synchronized stability of the protein–ligand complex.

On the other hand, the root-mean-square fluctuation (RMSF) analysis provides insight into the flexibility and fluctuation of individual residues within protein–ligand complexes during molecular dynamics simulations.

For the first complex, L1-1EAG (Figure 7), the results of the RMSF analysis reveal that residues exhibit limited fluctuations, with values not exceeding 2.8 Ǻ. Notably, the residues involved in interactions, highlighted in green in the figure, show particularly low fluctuations. This suggests that the interactions between the L1 ligand and the protein (1EAG) help to stabilize these specific residues, resulting in minimal structural variations throughout the simulation.

In the second complex, L13-3DJE (Figure 7), RMSF analysis indicates even smaller residue fluctuations, with values of no more than 2.4 Ǻ. This implies remarkable stability in the protein–ligand complex, where the L13 ligand interacts with the protein (3DJE). The reduced RMSF values mean that individual ligand and protein residues undergo minimal deviations from their mean positions, reinforcing the overall structural stability of the complex.

Collectively, these results suggest that protein–ligand interactions remain stable and that the system is resistant to significant structural perturbations throughout the simulation.

#### 2.4.2. Protein–Ligand Contact

Evaluation of the protein–ligand interaction fractions reveals essential details of the binding dynamics between the targeted proteins and the studied compounds. Specifically, compound L1 (Figure 8) showed interactions with 21 residues, while compound L13 (Figure 9) interacted with 28 residues. These interactions covered a variety of bond types, including hydrogen bonds, hydrophobic interactions, ionic interactions, and water bridges. The identification of key residues with substantial interaction fractions and a high number of contacts provides valuable insights into the specificity and importance of these molecular associations. For compound L13, critical residues such as TRP17, LYS53, VAL54, CYS216, CYS337, ALA338, and GLY359 emerged, highlighting the central role of these amino acids in facilitating compound binding to the target protein. Similarly, compound L1 demonstrated significant interactions with key residues, in particular ASP32, ASP120, and ASP218, further elucidating the complex network of interactions governing L1 binding affinity to the target protein. These results contribute to a comprehensive understanding of the molecular mechanisms underlying protein–ligand interactions during 100 nanoseconds.

The use of computational techniques holds great promise in the treatment of human diseases. These methods enable the rapid identification and optimization of potential therapeutic agents, significantly reducing the time and cost associated with drug discovery. As computational power and algorithms continue to improve, these techniques will play an increasingly vital role in developing personalized and targeted therapies.

## 3. Materials and Methods

### 3.1. Database Collection

In the present study, we explored the antifungal potential of various natural compounds. For this purpose, we utilized a phytotherapy database that includes Morocco’s rich botanical heritage and its deep-rooted traditional medicinal practices. This database, as established in previous research [15], includes phytoconstituents from eight principal Moroccan plant families: Lamiaceae, Rosaceae, zingiberaceae, Apiaceae, Asteraceae, Lauraceae, Fabaceae, and Myrtaceae, all of which have been historically recognized in Moroccan traditional medicine for their beneficial activities [28]. Our dataset comprises a total of 297 molecules from 27 medicinal plants within these families, as detailed in the previous section. A detailed list of these studied medicinal plants is provided in the Appendix A.

### 3.2. Molecular Docking Procedure

In our quest to identify potent antifungal agents, we undertook a comprehensive screening of 297 natural compounds derived from a variety of medicinal plants. This computational approach aims to predict the optimal positioning of a molecule relative to another, in a stable and robust complex [29]. We used fluconazole, a widely used triazole antifungal agent, as the reference drug [30]. Our primary focus was on two specific proteins: *Candida albicans* and *Aspergillus fumigatus* proteins; their 3D structures were uploaded from the Protein Data Bank (PDB) [31] under the PDB IDs 1EAG and 3DJE, respectively. Molecular docking, pivotal to our analysis, was executed using AutoDock 4.2 [32], which employed the Lamarckian Genetic Algorithm (LGA) for enhanced precision [33]. Before the initiation of docking, the protein structures were optimized using the Swiss PDB Viewer v4.1 [34], and the ligands underwent refinement via Avogadro software [35]. This dual optimization ensured the removal of water molecules, the incorporation of pivotal hydrogen atoms, and the recalibration of molecular charges, creating a favorable environment for protein–ligand interactions. A carefully configured grid box was positioned around each protein by using the AutoGrid tool (AutoGrid, Redwood City, CA, USA) to surround the protein-binding active site (Figure 10). With these settings in place, each of the 297 natural compounds, along with fluconazole, was docked against the target proteins using AutoDock Vina [36]. For each docking iteration, an output file (out. pdbqt) was generated, allowing for the in-depth visualization of binding interactions. Key interaction parameters, such as polar and non-polar interactions, hydrogen bond lengths, and specific amino acids that interacted with the compounds, were analyzed using Discovery Studio Visualizer [37].

### 3.3. Computational ADME and Drug-Likeness Analysis

Pharmacokinetic and pharmacodynamic properties are pivotal in evaluating the therapeutic viability of compounds [38]. For our selected compounds, these characteristics cover essential aspects such as absorption, distribution, metabolism, excretion, and toxicity (ADME-Tox). Utilizing the canonical SMILES, we engaged the Swiss ADME web server for an in-depth evaluation of drug-likeness, forecasting parameters such as lipophilicity, water solubility, and adherence to the drug-likeness rule of five [26]. Complementing this, the pkCSM server was employed to thoroughly assess ADME-Tox profiles, providing insights crucial for predicting and understanding the in vivo behavior of these compounds [39]. Together, these tools enabled a rigorous computational assessment, helping us to discover the compounds that align with both therapeutic potential and bioavailability norms while conforming to safety benchmarks.

### 3.4. Quantum Chemical Investigation

In our methodology, we first employed docking techniques to assess the binding affinities and interactions of a wide array of natural compounds with target proteins associated with *Candida albicans* and *Aspergillus fumigatus*. Post-docking, the compound showcasing the highest binding affinity was allocated for an in-depth quantum chemical analysis. Utilizing Density Functional Theory (DFT) within Gaussian software 6.0 [40], we optimized the geometry of this ranked compound with the B3LYP functional linked with the 6-31G(d) basis set [41]. This was followed by computing the electronic properties. Then, by incorporating the frontier molecular orbitals (FMOs) and the molecular electrostatic potential (MEP), we provided a deep understanding of the compound’s intrinsic reactivity as well as its expected reactivity upon binding to target proteins [42,43].

### 3.5. Dynamics Protocol

Molecular dynamics simulation (MDS) plays a critical role in understanding the nuances of molecular interactions [44]. In our study, we employed Desmond, a core component of the Schrödinger suite, to probe interactions between fungal targets and our most promising lead compound [45]. For a comparative framework, we also evaluated fluconazole, a well-established antifungal drug. All simulations were extended over 100-nanosecond intervals. The preparations were meticulously undertaken using Schrödinger’s Protein Preparation Wizard [45], ensuring a consistent pH of 7.4. The System Builder tool facilitated the design of a periodic simulation environment, deploying the TIP3P water model for solvation [46]. Ensuring system neutrality was pivotal, which was achieved via the incorporation of Na+/Cl− counter ions at a salt concentration of 0.15 M. The robust OPLS3e force field was instrumental in the system setup [47]. Upon equilibration, a free phase was executed under the NVT ensemble, wherein the temperature gradually increased and was subsequently maintained. Following this, an NPT equilibration was conducted, oriented by the Nose–Hoover thermostat (set at 300 K) and the isotropic Martyna–Tobias–Klein barostat (stabilized at 1.01325 bar) [48]. Subsequently, a molecular dynamics simulation unfolded over 100 nanoseconds. Throughout these simulations, we meticulously monitored system stability metrics, such as RMSD, RMSF, and protein–ligand contacts, with interaction fraction. The comprehensive insights derived from these simulations shed light on the binding affinities and efficacies of the compounds and fluconazole against the fungal targets.

## 4. Conclusions

This study highlights the urgent need to improve antifungal therapies using computational methodologies. Screening 297 natural compounds, we prioritized those with high target affinity and favorable ADME/T profiles as potential inhibitors. Notably, two natural compounds showed exceptional properties and binding scores, supported by DFT calculations and dynamic simulations that gave us a better understanding of fungal drug design. This hierarchical approach highlights our ability to identify and optimize natural molecules, paving the way for their integration into the development of potent antifungal drugs. In the future, our research will focus on rigorous experimental validation through in vitro and in vivo studies to elucidate their efficacy and mechanisms of action. In addition, we plan to expand our computational models to encompass a wider range of pathogens and resistance mechanisms. By combining computational insights with experimental validation, this study aims to advance the field of antifungal drug discovery, filling critical gaps in current treatment strategies.

## Figures and Tables

**Figure 1 pharmaceuticals-17-00886-f001:**
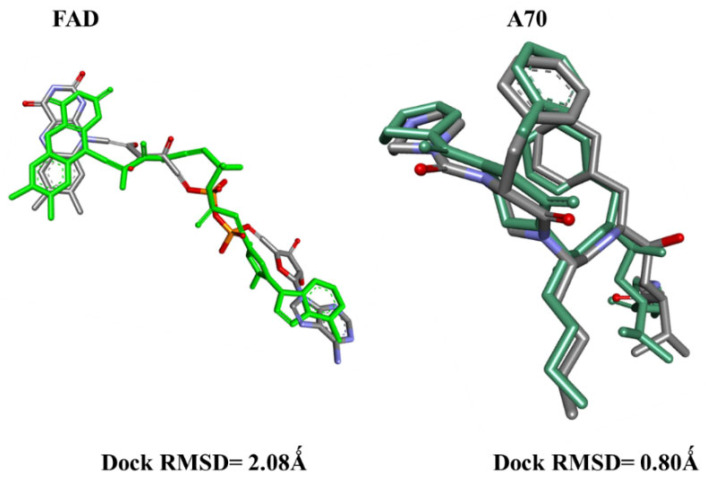
The superimposed poses of the original (gray) and redocked (green) FAD and A70 ligands inside the 1EAG and 3DJE receptor pockets.

**Figure 2 pharmaceuticals-17-00886-f002:**
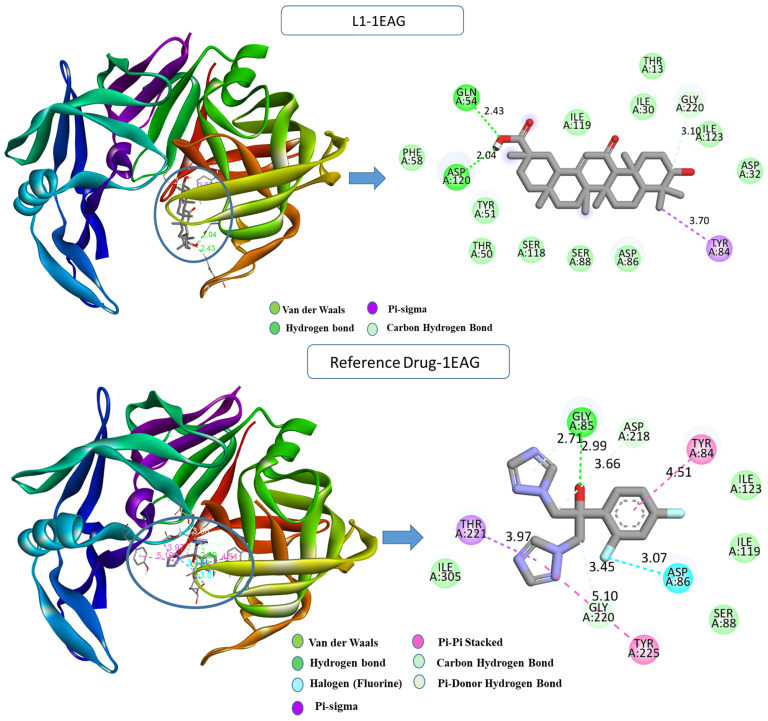
Two- and three-dimensional visualization of interactions between the compounds L1, the reference drug “fluconazole”, and the 1EAG protein target.

**Figure 3 pharmaceuticals-17-00886-f003:**
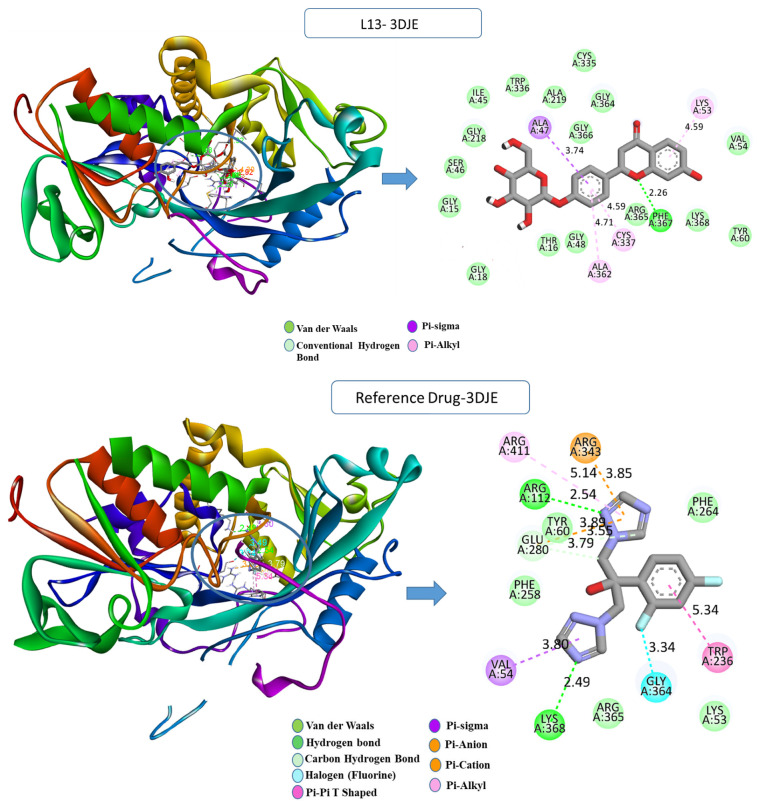
Two- and three-dimensional visualization of interactions between the compounds L13, the reference drug “fluconazole”, and the 3DJE protein target.

**Figure 4 pharmaceuticals-17-00886-f004:**
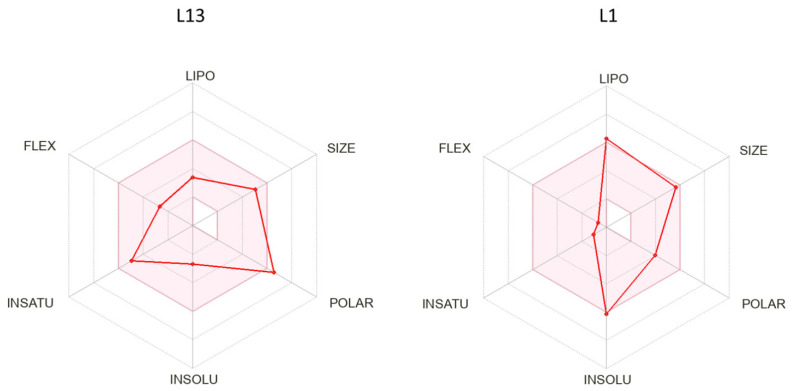
The predicted bioavailability radar of compounds L1 and L13.

**Figure 5 pharmaceuticals-17-00886-f005:**
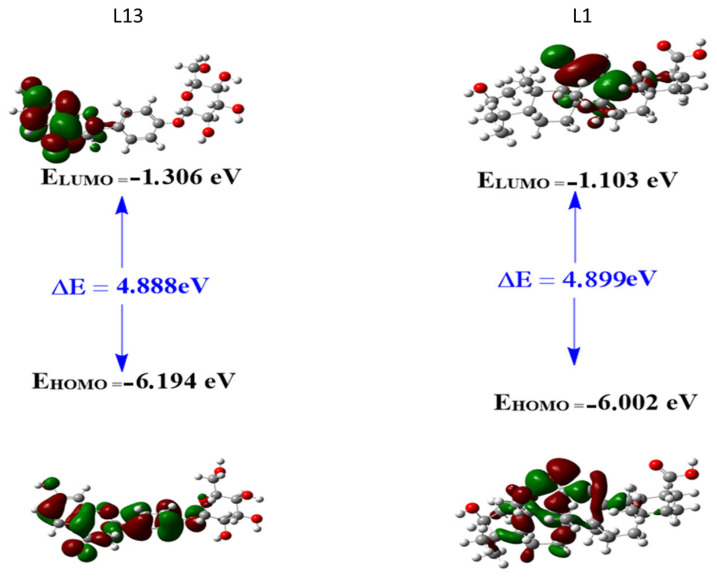
Frontier molecular orbitals (FMOs) for the studied compounds L1 and L13.

**Figure 6 pharmaceuticals-17-00886-f006:**
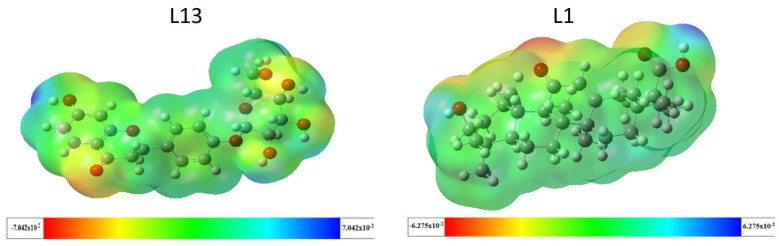
Electrostatic potential mapping for the compounds L13 and L1.

**Figure 7 pharmaceuticals-17-00886-f007:**
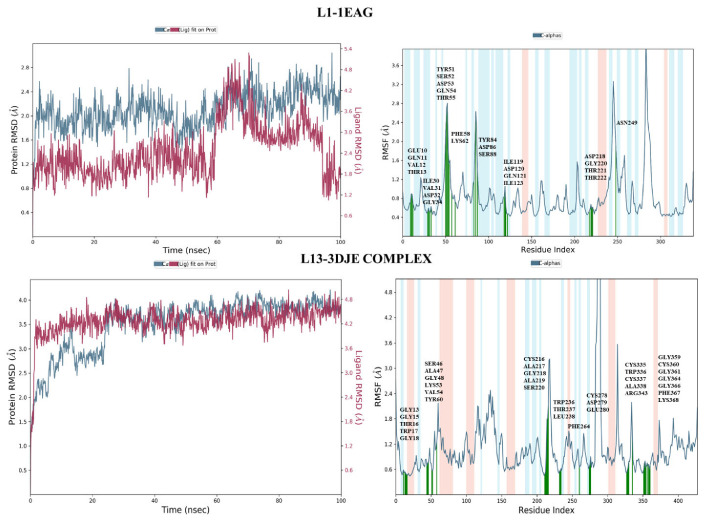
The RMSD and RMSF plots of both complexes L1-1EAG and L13-3DJE.

**Figure 8 pharmaceuticals-17-00886-f008:**
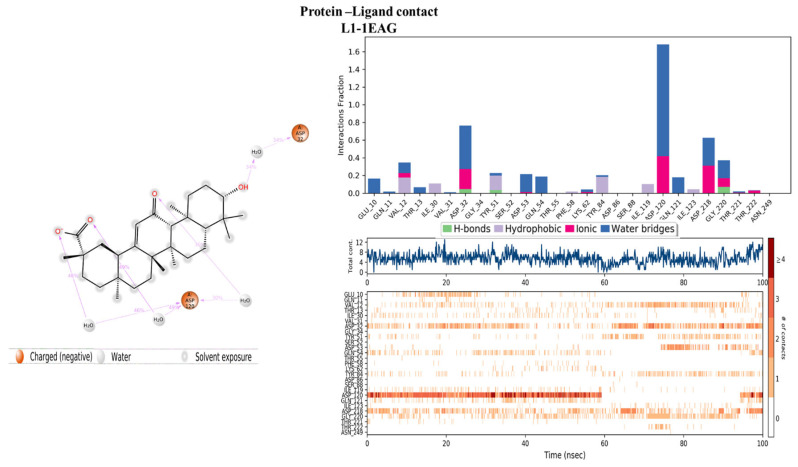
Protein–ligand histogram and number of contacts between compounds L1 and 1EAG protein.

**Figure 9 pharmaceuticals-17-00886-f009:**
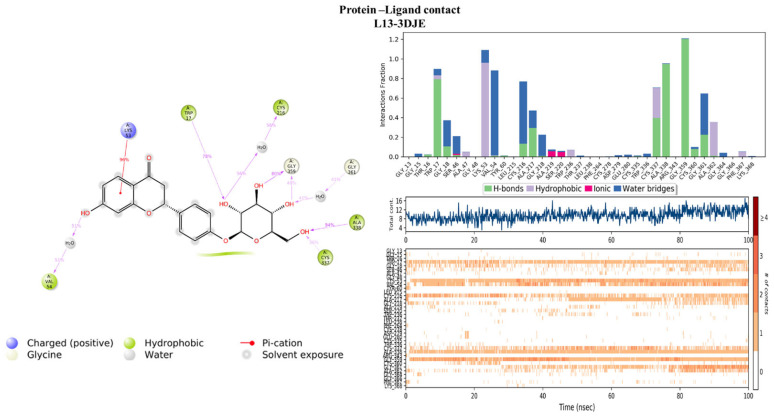
Protein–ligand histogram and number of contacts between compounds L13 and 3DJE protein.

**Figure 10 pharmaceuticals-17-00886-f010:**
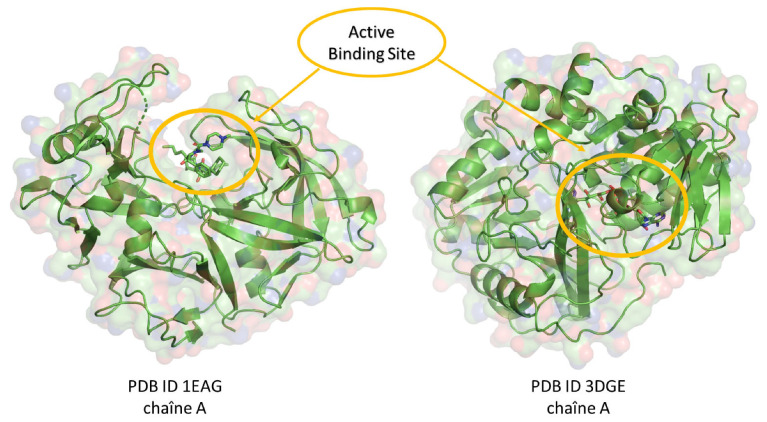
Three-dimensional representation of target proteins with well-defined active sites for protein–ligand–drug interaction.

**Table 1 pharmaceuticals-17-00886-t001:** Top-ranked 40 compounds among 297 natural compounds with corresponding binding energy scores (kcal/mol).

Ligand	Score (kcal/mol)		Ligand	Score (kcal/mol)		Ligand	Score (kcal/mol)		Ligand	Score (kcal/mol)	
	1EAG	3DJE		1EAG	3DJE		1EAG	3DJE		1EAG	3DJE
**L1**	−9.6	−8.3	**L11**	−8.3	−9.2	**L21**	−7.8	−8.6	**L31**	−7.4	−8.1
**L2**	−9.3	−9.6	**L12**	−8.3	−9	**L22**	−7.8	−9.2	**L32**	−7.4	−8.3
**L3**	−8.9	−8.5	**L13**	−8.3	**−10** **.** **1**	**L23**	−7.7	−8.8	**L33**	−7.4	−8.9
**L4**	−8.8	−8.4	**L14**	−8.2	**−10.8**	**L24**	−7.7	−9	**L34**	−7.4	−9.6
**L5**	−8.7	−9	**L15**	−8.2	−8.9	**L25**	−7.6	−9	**L35**	−7.3	−9.3
**L6**	−8.6	−9.9	**L16**	−8.2	−8.8	**L26**	−7.6	−8.8	**L36**	−7.3	−8.9
**L7**	−8.5	−8.4	**L17**	−8.1	−8.3	**L27**	−7.5	−8.4	**L37**	−7.2	−8.4
**L8**	−8.5	−8.6	**L18**	−8	−7.9	**L28**	−7.5	−9.4	**L38**	−7.1	−9.6
**L9**	−8.3	−8.7	**L19**	−7.9	−9.3	**L29**	−7.5	−8.5	**L39**	−7	−7.9
**L10**	−7.1	−8.7	**L20**	−7.8	−8.3	**L30**	−7.4	−8.5	**L40**	−7	−8.5
Scoring for the reference drug (kcal/mol)											
*Candida Albicans*/fluconazole		−6.7				*Aspergillus fumigatus*/fluconazole		−7.9			

**Table 2 pharmaceuticals-17-00886-t002:** Predicted bioavailability of compounds L1 and L13.

Molecule	L1	L13
MW (size)	470.68	418.39
GI absorption	High	Low
FractionCsp3 (insaturation)	0.87	0.38
#Rotatable bonds (flexibility)	1	4
#H-bond acceptors	4	9
#H-bond donors	2	5
TPSA (polarity)	74.60	145.91
XLOGP3 (lipophilicity)	5.49	0.39
MLOGP	4.87	−0.92
ESOL LogS (insolubility)	−6.15	−2.71
Lipinski #violations	1	0
Bioavailability score	0.85	0.55
PAINS #alerts	0	0
Synthetic accessibility	6.08	4.91

**Table 3 pharmaceuticals-17-00886-t003:** ADME-Tox profiles of both compounds L1 and L13.

ADMET	Properties	Compounds	
		L1	L13
Absorption	Water solubility (log mol/L)	−4.909	−3.669
	Caco2 permeability (log Papp in 10^−6^ cm/s)	0.912	0.371
	Intestinal absorption (human)%	97.389	56.977
	Skin permeability (log Kpp)	−2.713	−2.744
Distribution	VDss (human) (log L/kg)	−0.916	−0.070
	BBB permeability log BB	0.101	−1.167
	CNS permeability log PS	−1.320	−3.853
Metabolism	CYP2D6 substrate	No	No
	CYP3A4 substrate	Yes	Yes
	CYP1A2 inhibitor	No	No
	CYP2C19 inhibitor	No	No
	CYP2C9 inhibitor	No	No
	CYP2D6 inhibitor	No	No
	CYP3A4 inhibitor	No	No
Excretion	Total clearance (log mL/min/kg)	−0.114	0.734
Toxicity	AMES toxicity	No	No
	hERG I inhibitor	No	No
	Hepatotoxicity	No	No
	Skin sensitization	No	No

**Table 4 pharmaceuticals-17-00886-t004:** Quantum parameters for compounds L13 and L1 with their corresponding values.

Compound	L13	L1
E_LUMO_ (eV)	−1.3056	−1.1034
E_HOMO_ (eV)	−6.1941	−6.0022
ΔE	4.8885	4.8989
Χ	3.7498	3.5528
η	2.4443	2.4494
pi	−3.7499	−3.5528
σ	0.4091	0.4082
ω	2.876	2.577
Dipole moment µ (DEBYE)	1753.767	5735.663
Electronic energy	−40,558.4	−40,046.5

## Data Availability

The data are included in the article and Appendix A.

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
