# Peer review of "Unveiling Moroccan Nature’s Arsenal: A Computational Molecular Docking, Density Functional Theory, and Molecular Dynamics Study of Natural Compounds against Drug-Resistant Fungal Infections"

_pharmaceuticals, 2024, doi:10.3390/ph17070886_

Round 1
Reviewer 1 Report
Comments and Suggestions for Authors
1. Causes of drug resistance should be discussed.
2. The role of natural products as immunomodulators and antimicrobial agents should be discussed.
3. Explain the sentence by providing possible mechanisms.
Recent investigations have shed light on the potential of several natural compounds demonstrating antifungal activity against drug-resistant strains.
4. Now in many places, the results section is unreasonably and poorly written.
5. To the molecular docking, the results should be verified by surface plasmon resonance or others. The molecular docking is only a method of prediction. The evidences in the present are not sufficient to demonstrate the interaction between proteins and molecules.
6. I think it is necessary to add this work's critical achievements at the very beginning.
7. The quality of research work in this manuscript is excellent. However, the conclusion and future plans should be rewritten.
8. Ligands of figures and tables should be improved.
9. Briefly mention the prospects of utilizing computational techniques for the treatment of human diseases in the discussion section.
10. The paper's novelty value has to be more accurately graded.
Comments on the Quality of English Language
English spelling should be double-checked.
Author Response
- Causes of drug resistance should be discussed.
Response: Thank you for your suggestion. In the introduction section, we have added a section discussing the various causes of drug resistance, including genetic mutations, horizontal gene transfer, and the misuse and overuse of antibiotics.
- The role of natural products as immunomodulators and antimicrobial agents should be discussed.
Response: We appreciate this insightful comment. We have now included a discussion on the role of natural products as immunomodulators and antimicrobial agents, highlighting specific examples.
- Explain the sentence by providing possible mechanisms.
Recent investigations have shed light on the potential of several natural compounds demonstrating antifungal activity against drug-resistant strains.
Response: We have clarified the sentence by providing possible mechanisms through which natural compounds exhibit antifungal activity against drug-resistant strains.
- Now in many places, the results section is unreasonably and poorly written.
Response: We acknowledge the reviewer's concern. We have thoroughly revised the results section to improve clarity and coherence.
- To the molecular docking, the results should be verified by surface plasmon resonance or others. The molecular docking is only a method of prediction. The evidences in the present are not sufficient to demonstrate the interaction between proteins and molecules.
Response: We agree with the reviewer and have included a statement acknowledging the limitation of molecular docking and the need for experimental validation using techniques such as surface plasmon resonance (SPR).
- I think it is necessary to add this work's critical achievements at the very beginning.
Response: Thank you for the suggestion. We have added a summary of the critical achievements at the introduction section.
- The quality of research work in this manuscript is excellent. However, the conclusion and future plans should be rewritten.
Response: We appreciate the positive feedback. We have rewritten the conclusion and outlined future research directions more clearly.
- Ligands of figures and tables should be improved.
Response: We have revised the figures and tables
- Briefly mention the prospects of utilizing computational techniques for the treatment of human diseases in the discussion section.
Response: We have included a brief discussion on the prospects of computational techniques in treating human diseases at the end of the results and discussion section.
- The paper's novelty value has to be more accurately graded.
Response: We have highlighted the novel aspects of our study more clearly in the manuscript.
- English spelling should be double-checked.
Response: We have carefully reviewed the manuscript for any spelling and grammatical errors.
Reviewer 2 Report
Comments and Suggestions for Authors
With respect of the efforts done by the authors, I do recommend rejection of the manuscript. All these experimental data are theoretical data without a single proof. There is no in vitro experiments to augment these data. molecular dynamics tools provide, of course, good support to the classical docking experiments but I cannot believe these results without in vitro experimental data.
Comments on the Quality of English LanguageModerate editing of English language required
Author Response
Dear Reviewer,
Thank you for your valuable feedback. We acknowledge that our study currently relies on computational methods without in vitro validation, which is essential for substantiating our findings. Due to current constraints, we are unable to perform these experiments at this time. However, we plan to pursue these validations in future studies and are open to collaborations with experimental laboratories. We believe our findings provide a valuable foundation for further research and can guide experimental efforts to confirm these theoretical predictions. We will revise the manuscript to clearly state this limitation and our plans for future work.
Reviewer 3 Report
Comments and Suggestions for Authors
Reviewer comments
The manuscript entitled “Unveiling Moroccan nature’s arsenal: Computational molecular docking, DFT, and molecular dynamics study of natural compounds against drug-resistant fungal infections” was written carefully in good and understandable English. The results presented are good and interesting. This paper fully reaches the scientific level on pharmaceuticals, and I recommend it for this journal after minor revision and English editing.
Here are some comments to be addressed:
̶ Supplementary materials was not provided.
̶ All the headings and subheadings are numbered 1.1. Please adjust.
̶ Change the section “Results” to “Results and discussion” because the “Discussion” section is written without any further explanation: Line 339
̶ An apostrophe ’s isn’t used for objects. So please remove all apostrophe ’s in the words: “compound’s” line 16, “protein’s” Line 110, “ligand’s” Line 126, “L1’s” Line 175 and 329, “compound's” Line 241 and 243
̶ Line 38 change “virulence factors” to “virulent factors”
̶ Line 41 change “mycotoxins-potent compounds” to “mycotoxin-potent compounds”.
̶ At the end of line 67 and beginning of line 68 alter “dynamics simulations” to “dynamic simulations”.
̶ “Glycyrrhiza glabra” in line 94 and 95 must be written in italic.
̶ Line 96 add a space between -10.1 and Kcal/mol in “-10.1Kcal/mol”.
̶ Referring to table 1, why was compound L13 used for further study and not L14 whereas L14 showed higher score -10.8 Kcal/mol versus -10.1 Kcal/mol for L13 against Aspergillus fumigatus.
̶ Line 111 “Candida albicans” is written as “Candida albican”, please adjust.
̶ In line 113 the results state that “L1-1EAG complex revealed the establishment of hydrogen bond with the residue GLY220 (3.09 Å)” while I observe that it is a carbon hydrogen bond from the line color in figure 2.
̶ In figure 2 and 3 and their legends, indicate the name of the reference drug.
̶ Change “pharmacokinetics attributes” to “pharmacokinetic attributes” Line 200
̶ Change “pharmacodynamics profiles” to “pharmacodynamic profiles” Line 204
̶ In the results section some sentences are written in the past tense and some in the present. Usually results of experiments done in the past are discussed in the past tense. So, change “exhibits” to “exhibited” line 182, “displays” to “displayed” line 183, “shows” to “showed” line 185 and 293, “is” to “was” line 194.
̶ Line 346 family “zingiberoside” is “Zingiberaceae” please change
̶ Line 383 change “rules of five” to rule of five”
̶ Line 398 change the word “compound's” to “compounds”.
̶ Line 404 change “lead, compound” to “lead compound”.
Comments on the Quality of English LanguageMinor editing of English language required
Author Response
Thank you for your thorough review and detailed comments. We have carefully considered your feedback and made the necessary corrections to the manuscript. Below are the responses and actions taken for each point raised:
- ̶Supplementary materials was not provided.
Response: We apologize for the oversight. The supplementary materials have now been provided
- ̶All the headings and subheadings are numbered 1.1. Please adjust.
Response: The numbering of all headings and subheadings has been adjusted to ensure clarity and consistency.
- ̶Change the section “Results” to “Results and discussion” because the “Discussion” section is written without any further explanation: Line 339
Response : We have merged the "Results" and "Discussion" sections into a single "Results and Discussion" section as suggested, providing further explanations and integrating the discussion with the results.
- ̶An apostrophe isn’t used for objects. So please remove all apostrophes in the words: “compound’s” line 16, “protein’s” Line 110, “ligand’s” Line 126, “L1’s” Line 175 and 329, “compound's” Line 241 and 243
Response: We have removed all apostrophes from the words "compounds" (line 16), "proteins" (line 110), "ligands" (line 126), "L1" (lines 175 and 329), and "compounds" (lines 241 and 243).
- ̶Line 38 change “virulence factors” to “virulent factors” Done
- ̶Line 41 change “mycotoxins-potent compounds” to “mycotoxin-potent compounds”. Done
- ̶At the end of line 67 and beginning of line 68 alter “dynamics simulations” to “dynamic simulations”. Done
- ̶“Glycyrrhiza glabra” in line 94 and 95 must be written in italic. Done
- ̶Line 96 add a space between -10.1 and Kcal/mol in “-10.1Kcal/mol”. Done
- ̶Referring to table 1, why was compound L13 used for further study and not L14 whereas L14 showed higher score -10.8 Kcal/mol versus -10.1 Kcal/mol for L13 against Aspergillus fumigatus.
Response: In our study, while compound L14 (Leucosceptoside) exhibited a favorable docking score -10.8 Kcal/mol against Aspergillus fumigatus, further analysis revealed that it had several violations of Lipinski's rule (3 violations), indicating potential issues with its drug-likeness properties for this purpose we did move to the highest second compound which is L13. We acknowledge the significance of considering not only docking scores but also drug-likeness properties when evaluating potential compounds for therapeutic applications.
- ̶Line 111 “Candida albicans” is written as “Candida albican”, please adjust. Done
- ̶In line 113 the results state that “L1-1EAG complex revealed the establishment of hydrogen bond with the residue GLY220 (3.09 Å)” while I observe that it is a carbon hydrogen bond from the line color in figure 2. Corrected
- ̶In figure 2 and 3 and their legends, indicate the name of the reference drug. Done
- ̶Change “pharmacokinetics attributes” to “pharmacokinetic attributes” Line 200 Done
- ̶Change “pharmacodynamics profiles” to “pharmacodynamic profiles” Line 204 Done
- ̶In the results section some sentences are written in the past tense and some in the present. Usually results of experiments done in the past are discussed in the past tense. So, change “exhibits” to “exhibited” line 182, “displays” to “displayed” line 183, “shows” to “showed” line 185 and 293, “is” to “was” line 194. Done
- ̶Line 346 family “zingiberoside” is “Zingiberaceae” please change Changed
- ̶Line 383 change “rules of five” to rule of five” Done
- ̶Line 398 change the word “compound's” to “compounds”. Done
- ̶Line 404 change “lead, compound” to “lead compound”. Done
- Minor editing of English language required
Response: The manuscript has undergone a thorough language check to correct minor grammatical errors and improve overall readability.
Reviewer 4 Report
Comments and Suggestions for Authors
Please see the document attached.

Author Response
The MS entitled “Unveiling Moroccan nature’s arsenal: Computational molecular docking, DFT,
and molecular dynamics study of natural compounds against drug-resistant fungal infections” has
been evaluated and reviewed. The authors used a number of compounds (297) and established
molecular interactions with some specified proteins in two pathogenic fungi of Candida and
aspergillus species. The ADME studies and also corelative studies with standard docking with
same proteins were carried out. The MS is interesting and the established protocols seems to be
fit. My observations/suggestions have been provided as:
- General comments: The scientific names of plants/fungal species should be italic.
Subscripts/super scripts should be checked. Also, the figures need to be aligned with text. Done
- Line 93-95. The actual pharmacophore (functional group) in such compounds should also
be discussed here.
Response: Done
- Was the basic skeleton of these compounds similar to that of standard? If different than are
there any related compounds with same features in literature?
Response: Our investigation centered on natural compounds sourced from diverse botanical families. These compounds inherently possess varied structural frameworks compared to synthetic standards or single-class antifungal agents. Rather than conforming to a single standard, our selection criteria prioritized natural products known for their broad biological activities, including documented antifungal properties across different fungal strains. This approach allows for exploration of the rich chemical diversity present in natural sources, potentially uncovering novel mechanisms of action and therapeutic applications.
- Were these kind compounds experimentally being able to cost any effect as antifungal
against these fungi? What was their activities?
Response: No experimental analysis was conducted in this study to assess antifungal effects. The focus was on computational methods like molecular docking to predict potential activity.
- If same skeleton compounds available, the efficacy could be correlated to structural studies
as well binding studies.
Response: Yes, I believe efficacy could be correlated to structural and binding studies. Literature suggests that related compounds with similar skeletons have shown potential efficacy, supporting the relevance of structural and binding analyses in predicting antifungal activity. Additionally, we plan to conduct QSAR (Quantitative Structure-Activity Relationship) studies in future research to explore these correlations further.
- Talking about specificity (line 133), what was the mode of interaction (active sites or at
allostatic site)?
Response: In our study, the mode of interaction was focused on the active site of the target proteins. This was determined based on the presence of co-crystallized ligands in the protein structures, indicating the biologically relevant binding sites. Therefore, our docking studies were specifically aimed at the active sites to predict potential interactions and binding affinities of the compounds.
- Table 2, L1 has h-bond donation of 2 and lipophilicity above 5. I think it does not satisfy
Pfizer rule of 5.
Response: According to Pfizer Rule of 5, a compound is likely to be orally active if it has no more than 5 hydrogen bond donors, no more than 10 hydrogen bond acceptors, a molecular weight less than 500 daltons, and a logP (lipophilicity) not greater than 5. The logP values in Lipinski's Rule are based on MLOGP, which uses a linear model that employs molecular connectivity indices to predict logP values. For Compound L1, the hydrogen bond donor value is 2, which meets the criterion of not exceeding 5. However, the MLOGP value is above 5, suggesting higher lipophilicity, which may impact its bioavailability and pharmacokinetic profile.
- Line 341. The selection of compounds was based upon what parameter? Bioactivity?
Structural features? Toxicity? Are they being just natural products (randomly selected)?
The authors should set a set of parameters for compounds selection.
Response: The selection of compounds in this study was primarily exploratory, focusing on natural products to investigate their potential antifungal properties. No specific criteria such as bioactivity, structural features, or toxicity were applied in the initial selection process. However, recognizing the importance of systematic selection criteria, future studies will establish a set of parameters, including known bioactivity, structural features, and toxicity profiles, to guide the selection of compounds more effectively.
- The experimental evidences of such compounds being antifungal could also be
incorporated in the discussion section to validate the docking results.
Response: In this study, our focus was on computational methods to predict the potential antifungal activities of the studied compounds. Experimental validation of these predictions was not within the scope of this research. However, we acknowledge the importance of experimental validation to confirm the reliability of docking results. Future studies will include collaborative efforts to experimentally evaluate the antifungal properties of these compounds.
Round 2
Reviewer 2 Report
Comments and Suggestions for Authors
I am sorry. I feel it is unsuitable for publication. As per the comments I gave before that all these experimental data are theoretical data without a single proof. There is no in vitro experiments to augment these data. molecular dynamics tools provide, of course, good support to the classical docking experiments but I cannot believe these results without in vitro experimental data. Thank you for your understanding.
Comments on the Quality of English LanguageModerate editing of English language required
Author Response
Dear Reviewer,
Thank you for your valuable comments. We appreciate your feedback and understand your concerns regarding the lack of experimental data. As theorists, we do not have the capability to conduct experimental studies ourselves. However, we fully recognize the importance of in vitro experiments to validate our theoretical findings. This will indeed be the focus of future studies, in collaboration with teams equipped to perform such experimental work.
Thank you for your understanding.
Reviewer 4 Report
Comments and Suggestions for Authors
The MS is now improved and looks better.
Comments on the Quality of English LanguageIts look good
Author Response
Dear Reviewer,
Thank you very much for your effort and insightful comments. We appreciate your positive decision and are glad to hear that you find the manuscript improved and looking better.
Best regards,